# Addressing the Elephant in the Immunotherapy Room: Effector T-Cell Priming versus Depletion of Regulatory T-Cells by Anti-CTLA-4 Therapy

**DOI:** 10.3390/cancers14061580

**Published:** 2022-03-20

**Authors:** Megan M Y Hong, Saman Maleki Vareki

**Affiliations:** 1Department of Pathology and Laboratory Medicine, University of Western Ontario, London, ON N6A 3K7, Canada; mhong32@uwo.ca; 2London Regional Cancer Program, Lawson Health Research Institute, London, ON N6A 5W9, Canada; 3Department of Oncology, University of Western Ontario, London, ON N6A 3K7, Canada; 4Department of Medical Biophysics, University of Western Ontario, London, ON N6A 3K7, Canada

**Keywords:** anti-CTLA-4, CTLA-4, Tregs, FOXP3, CD28 costimulation, immunotherapy, ADCC/P, ipilimumab

## Abstract

**Simple Summary:**

Immunotherapy has transformed the treatment of advanced cancers by leveraging patients’ immune system in attacking tumor cells. Cytotoxic T-lymphocyte Associated Protein 4 (CTLA-4) is an immune checkpoint molecule highly expressed on regulatory T-cells (Tregs) that suppresses the function of tumor-reactive T-cells responsible for eliminating cancer cells. CTLA-4-targeting agents can release the “brakes” off the immune system to promote anti-tumor immune responses. Anti-CTLA-4 has been used to treat various solid cancers for over a decade; however, its exact mechanism(s) of action is still widely debated. There is a need to understand the fundamental mechanism(s) of anti-CTLA-4 function to employ strategies to improve its therapeutic efficacy and develop the next-generation of anti-CTLA-4 antibodies. This article is an in-depth analysis of the proposed mechanisms of anti-CTLA-4 therapy and its newfound uses in cancer treatment.

**Abstract:**

Cytotoxic T-lymphocyte Associated Protein 4 (CTLA-4) is an immune checkpoint molecule highly expressed on regulatory T-cells (Tregs) that can inhibit the activation of effector T-cells. Anti-CTLA-4 therapy can confer long-lasting clinical benefits in cancer patients as a single agent or in combination with other immunotherapy agents. However, patient response rates to anti-CTLA-4 are relatively low, and a high percentage of patients experience severe immune-related adverse events. Clinical use of anti-CTLA-4 has regained interest in recent years; however, the mechanism(s) of anti-CTLA-4 is not well understood. Although activating T-cells is regarded as the primary anti-tumor mechanism of anti-CTLA-4 therapies, mounting evidence in the literature suggests targeting intra-tumoral Tregs as the primary mechanism of action of these agents. Tregs in the tumor microenvironment can suppress the host anti-tumor immune responses through several cell contact-dependent and -independent mechanisms. Anti-CTLA-4 therapy can enhance the priming of T-cells by blockading CD80/86-CTLA-4 interactions or depleting Tregs through antibody-dependent cellular cytotoxicity and phagocytosis. This review will discuss proposed fundamental mechanisms of anti-CTLA-4 therapy, novel uses of anti-CTLA-4 in cancer treatment and approaches to improve the therapeutic efficacy of anti-CTLA-4.

## 1. Introduction

Cytotoxic T-Lymphocyte Associated Protein 4 (CTLA-4) is an immune checkpoint molecule expressed by T-cells. CTLA-4 blockade was first shown to reject tumors and inhibit the growth of established tumors in preclinical models. Furthermore, an immunological memory response was formed with CTLA-4 blockade on cured animals as tumors were rejected upon secondary challenge [1]. The discovery of the role of anti-CTLA-4 therapy on anti-tumor immunity ultimately led to the first and only FDA approval of anti-CTLA-4 immunotherapy to date, ipilimumab, in 2011. Ipilimumab was first approved for the treatment of unresectable advanced melanoma and later approved as adjuvant therapy for resectable stage III melanoma in 2015 [2,3]. Ipilimumab monotherapy in a phase III trial in unresectable stage III and IV melanoma patients had the best overall response rate (10.9%) compared to ipilimumab plus glycoprotein 100 (gp100) vaccine (5.7%) or gp100 monotherapy (1.5%). Sixty percent of patients in the ipilimumab monotherapy group maintained an objective response for at least two years. However, immune-related adverse events (irAEs) occurred in ~60% of patients on either ipilimumab treatment arm [2]. 

To date, there is still a lack of clear understanding of why only a fraction of patients respond to anti-CTLA-4 therapy, and the exact mechanism(s) of anti-CTLA-4 remain elusive. CTLA-4 blockade on conventional T-cells and regulatory T-cells (Tregs) results in maximal anti-tumor activity compared to unicompartmental blockade of either T-cell population [4]. However, preclinical evidence suggests that depletion of Tregs or abrogating their function may be the main contributor to the anti-tumor activity of CTLA-4 blockade [4]. This review will explore proposed mechanisms of anti-CTLA-4 involving Treg inhibition and highlights novel approaches to improve the efficacy of anti-CTLA-4 therapy while reducing its toxicity in cancer treatment.

## 2. Tregs and CTLA-4 in Cancer

Conventional Tregs are CD4^+^CD25^+^FOXP3^+^ T-cells that suppress excessive innate and adaptive immune responses to promote self-tolerance and tissue homeostasis [5,6,7]. Despite only making up 1–10% of CD4^+^ lymphocytes in the peripheral blood, selective depletion of Tregs in mice leads to a scurfy-like phenotype characterized by lymphadenopathy, splenomegaly, skin inflammation, and insulitis [8,9,10,11]. Similarly, dysfunctional Tregs can lead to autoimmune conditions in humans, and mutations in FOXP3—a signature transcription factor of Tregs—can lead to immune dysregulation polyendocrinopathy enteropathy X-linked (IPEX) syndrome [12]. 

Tumors often recruit Tregs to the tumor microenvironment (TME) and utilize their immunosuppressive functions to evade the host’s anti-tumor immune responses [13,14,15,16,17,18,19,20]. The Treg levels in the TME and peripheral blood of cancer patients are often higher than healthy controls and are correlated with poor clinical outcomes and lower overall survival [21,22,23,24,25,26]. In addition, intra-tumoral Tregs exhibit a greater suppressive capacity than peripheral Tregs and differ in the expression of immunosuppressive markers [27,28]. Tregs can facilitate immunosuppression through several cell contact-dependent and -independent mechanisms that make it a viable target for immunotherapy. For example, Tregs express several immune checkpoint molecules such as Programmed Cell Death-1 (PD-1), CTLA-4, Lymphocyte-Activation Gene 3 (LAG-3), and T-cell Immunoglobulin-and Mucin-Domain Containing Protein 3 (TIM-3) that can inhibit the activation of effector T-cells [7,25,29]. Moreover, IL-10, IL-35, and TGF-β secreted by Tregs can inhibit T-cell activation, differentiation, cytokine production, and induce effector T-cell exhaustion [7]. 

A hallmark study conducted by Shimizu et al. showed that depletion of Tregs can increase the generation of cytotoxic T-cells leading to tumor eradication and increased survival in several mouse models [30]. In addition, the efficacy of immune checkpoint inhibitors (ICIs) can be enhanced by promoting immune cell infiltration and effector T-cell functions. Therefore, inhibition of intra-tumoral Tregs may be critical for treating immunologically-cold tumors that respond poorly to ICIs [31,32]. 

CTLA-4 is constitutively expressed on Tregs, whereas conventional T-cells express it transiently after activation. CTLA-4 prevents costimulatory signaling required for T-cell activation by competing with CD28 on conventional T-cells in binding to CD80/86 on antigen-presenting cells (APCs) [7,33]. Moreover, CTLA-4 is critical for the immunosuppressive capacity of Tregs and immunoregulation [34,35]. In vivo studies show that CTLA-4 expression in the effector T-cell compartment alone is insufficient for immunosuppression as selective knock-out of CTLA-4 on Tregs causes lymphadenopathy, splenomegaly, myocarditis, and gastritis in mice [36,37]. Due to its constitutive expression on Tregs, CTLA-4 is an ideal target to selectively deplete or inhibit the suppressive functions of Tregs in the TME.

## 3. Blockading CD80/86-CTLA-4 Interactions

Enhancement of effector T-cell activation by blockading CD80/86-CTLA-4 interactions was the first proposed mechanism of anti-CTLA-4 antibodies (Table 1) [38]. T-cell receptor (TCR) engagement with antigen-presenting MHC molecules along with costimulatory signaling through CD28 and CD80/86 engagement is crucial for T-cell activation and proliferation [33]. CTLA-4 can block costimulatory signaling as it has a higher affinity for CD80/86 than CD28 (Figure 1) [33]. CD80/86-CTLA-4 engagement does not induce apoptosis in T-cells but inhibits T-cell proliferation by decreasing the production of IL-2 and the expression of IL-2Rα (CD25) on activated T-cells [39,40]. In addition, CD28-CD80/86 interactions are required for conventional T-cells to upregulate aerobic glycolysis to facilitate the production of glycolytic intermediates and energy used to support cell division. CD28 costimulation increases glucose uptake and glycolysis in activated T-cells by increasing the expression of GLUT1 and activation of the PI3K/AKT and mTOR signaling pathways [41,42]. Combined blockade of PD-1 and CTLA-4 is effective against melanoma and renal cell carcinoma as it can enhance the effector function of CD8^+^ tumor-infiltrating lymphocytes (TILs) by increasing CD28 costimulation. CD28 costimulation increases glycolysis and mitochondrial function in CD8^+^ TILs and is accompanied by increased activation, proliferation, and production of effector molecules including granzyme B, IFN-γ, and TNF-α [42]. Expansion of TILs increases the T_effector_: Treg ratio in favor of tumor rejection by shifting an immunosuppressive TME to an immune-inflamed phenotype that is positively correlated with improved prognosis in patients [29,43,44]. 

Inducible T-cell costimulator (ICOS) is a member of the CD28 superfamily and is often upregulated on activated T-cells. ICOS binds to ICOS ligand (ICOSL) expressed on APCs that is also part of the CD80/86 superfamily. Similar to CD28, ICOS induces T-cell proliferation and cytokine production, especially in CD4^+^ T-cells. ICOS differs from CD28 as it does not compete with CTLA-4 in receptor-ligand interactions, but CTLA-4 engagement with CD80/86 inhibits the upregulation of ICOS [45]. ICOS expression has been implicated in enhancing pro-inflammatory immune responses by enhancing T-cell survival, Th2 cytokine production (IL-4, IL-5, IL-13), and IL-2 production required for early T-cell proliferation that can be abrogated by CTLA-4 engagement [45,46]. However, ICOS may also enhance the suppressive capacity of Tregs by increasing IL-10 and TGF-β production, FOXP3 expression, proliferation, and survival [47]. 

While ICOS has a dual role in autoimmunity and immunosuppression, it is essential in anti-tumor immune responses. The ICOS pathway may have a more prominent role in anti-tumor activity than immunosuppression as ICOS/ICOSL-deficient mice show impaired anti-tumor activity in response to anti-CTLA-4 [48]. The efficacy of anti-CTLA-4 can be enhanced when the ICOS pathway is simultaneously activated in melanoma and prostate cancer mouse models [49]. Concurrent ICOS pathway engagement further suppressed tumor growth, established a more robust memory response than anti-CTLA-4 alone, and increased survival in those models [48,49]. In addition, anti-CTLA-4 increases the intra-tumoral T_effector_: Treg ratio by promoting the expansion of ICOS^+^ effector T-cells in favor of anti-tumor activity (Figure 1) [48]. Furthermore, tumor-specific ICOS^hi+^CD4^+^IFNγ^+^ T-cells have been identified as a potential biomarker for anti-CTLA-4 response and prognosis in melanoma and bladder cancer patients [50,51]. 

Anti-CTLA-4 monoclonal antibodies (mAbs) and Fab fragments have been shown to enhance polyclonal or antigen-specific T-cell activation and cytokine production in CD80/86-dependent stimulation assays [52]. However, recent studies have shown that inhibition of CD80/86-CTLA-4 engagement may not be necessary for the efficacy of anti-CTLA-4. The ability of anti-CTLA-4 to block CD80/86-CTLA-4 engagement is clone dependent and assay dependent. Results using competitive ELISAs and cell-based assays to study the ability of anti-CTLA-4 to block CD80/86-CTLA-4 interactions differ depending on whether CD80/86 or CTLA-4 is immobilized in the assay. Ipilimumab, 9H10, and other anti-CTLA-4 clones can efficiently block plate-bound CTLA-4 from interacting with soluble CD80/86 [53,54]. However, Du et al. showed that ipilimumab and 9H10 fail to inhibit CD80/86-CTLA-4 interactions with plate-bound CD80/86 and soluble CTLA-4 [54]. The latter may be more physiologically relevant as CD80/86 are functional as transmembrane proteins [55]. Although CTLA-4 is primarily expressed as a transmembrane protein, functionally native soluble forms have been identified in human sera [56]. Moreover, ipilimumab failed to disrupt pre-existing CD80/86-CTLA-4 complexes and inhibit CD80/86-CTLA-4 mediated cell-cell contact. However, ipilimumab decreased the absolute number of Tregs, increased effector T-cell activation, and the T_effector_: Treg ratio in MC38 murine colon carcinoma tumors [54]. Previously, X-ray crystallography showed that ipilimumab binds to an interface of CTLA-4 that can blockade interactions with CD80/86 [57,58]. However, this evidence suggests that anti-tumor activity induced by anti-CTLA-4 is not mediated by inhibiting CD80/86-CTLA-4 interactions. Compared to ipilimumab, other anti-CTLA-4 mAbs that sufficiently block CD80/86-CTLA-4 interactions or known mAbs unable to block these interactions similarly rejected MC38 and B16-F10 tumors. Furthermore, adding CD80/86 neutralizing mAbs with ipilimumab did not affect MC38 tumor rejection. Du et al. showed that abrogation of CD80/86-dependent costimulation might not be an immunotherapeutic mechanism of ipilimumab even at high doses. They suggest that Treg depletion and tumor rejection by anti-CTLA-4 is instead Fc-receptor mediated [54]. However, Vandenborre et al. showed that humanized anti-CTLA-4 Fab fragments sufficiently increase T-cell activation and cytotoxic functions in mixed lymphocyte reaction assays [52]. Evidence to date suggests that concomitant Fc-mediated depletion of Tregs alongside blockading CD80/86-CTLA-4 interactions may be necessary for maximal anti-tumor activity [28,59].

## 4. Anti-CTLA-4-Mediated Antibody-Dependent Cellular Cytotoxicity and Phagocytosis

Antibody-dependent cellular cytotoxicity and phagocytosis (ADCC, ADCP) is targeted killing of antibody opsonized target cells by innate cells that express Fc receptors (FcRs). ADCC/P is mediated by natural killer (NK) cells, monocytes/macrophages, and polymorphonuclear leukocytes. Upon binding to FcRs, innate cells can phagocytose target cells or secrete perforin, granzymes, IFN-γ, and upregulate FAS ligand to induce apoptosis in target cells [60]. Ipilimumab has been demonstrated to deplete tumor-resident Tregs through ADCC/P in vivo and in vitro [53,61,62,63]. Ipilimumab has an IgG1 isotype that has an affinity towards FcγRI (CD64), FcγRIIa/b (CD32a/b), and FcγRIIIa (CD16a) [28,63]. In humans, NK cells mainly express FcγRIIIa, whereas monocytes/macrophages express FcγRI, FcγRIIa/b, and FcγRIIIa [64]. Intra-tumoral CD56^+^ NK cells, CD14^+^CD16^++^ non-classical monocytes, and CD68^+^CD14^+^ macrophages have been positively correlated with ipilimumab response in melanoma patients [28,62,65,66]. Human CD14^+^CD16^++^ nonclassical monocytes, but not CD14^+^CD16^-^ monocytes, were demonstrated to lyse Tregs in vitro in the presence of ipilimumab (Figure 2) [62].

NK cell activation markers and response to anti-CTLA-4 therapy have been positively correlated in melanoma patients. However, the direct involvement of NK cells in the therapeutic effect of anti-CTLA-4 is not well understood [65,66]. Pseudotime trajectory analysis on scRNA-seq data from intra-tumoral NK cells isolated from mouse sarcomas revealed that the expression of perforin and granzymes were induced by anti-CTLA-4 treatment (Figure 2) [66]. Additionally, intra-tumoral depletion of Tregs in CT26 colon carcinoma tumor-bearing mice was positively correlated with CD107a expression on intra-tumoral NK cells after anti-CTLA-4 treatment [65]. Nevertheless, the involvement of NK cells in the therapeutic effects of anti-CTLA-4 is widely debated, as NK cell depletion in mice does not affect the depletion of intra-tumoral Tregs [63]. In mice, FcγRIV and FcγRIII are orthologs of FcγRIIIa and FcγRIIa in humans, respectively, that mediate ADCC/P. NK cells in mice primarily express FcγRIII, whereas monocytes/macrophages express FcγRIIb, FcγRIII, and FcγRIV [64]. Similar to humans, anti-CTLA-4 mediated intra-tumoral Treg depletion in mice is associated with CD11b^+^FcγRIV^+^ macrophages. Treg depletion was abrogated in FcγRIV^−/−^ mice, but not in FcγRIII^−/−^ mice or NK cell-depleted wild-type mice [63]. However, NK cells may have a more prominent role in the effects of anti-CTLA-4 in humans as they express FcγRIIIa that can facilitate ADCC/P [64]. In mice, the lack of NK cell involvement may result from low FcγRIV expression on peripheral and intra-tumoral NK cells compared to monocytes [65]. Although human FcγRIIIa and its ortholog in mice play a similar role in anti-CTLA-4-mediated Treg depletion, it highlights the importance of animal models with humanized FcRs in studying Fc-dependent therapeutics on a cellular level.

The efficiency of host FcRs to facilitate ADCC/P and the Fc domain contribute to the efficacy of anti-CTLA-4 antibodies. CD16-V158F SNPs that confer a higher binding affinity to IgG1 antibodies are positively associated with the clinical response to Fc-dependent therapeutics. Metanalysis studies on advanced melanoma patients with high neoantigen burden revealed that CD16-V158F SNPs were associated with higher response rates to ipilimumab [28]. Furthermore, ipilimumab reduced tumor burden in human *CTLA4* knock-in mice transplanted with MC38 tumors, but this effect was inhibited with the co-administration of anti-FcR [53,54]. Similarly, impairing FcR engagement by inducing N297Q mutation to the Fc domain of ipilimumab abrogated the anti-tumor effect where tumor burden was comparable to IgG isotype controls [53]. Fc-dependent therapeutic efficacy of anti-CTLA-4 has similarly been demonstrated by several groups that observed reduced anti-tumor activity and Treg depletion when the Fc-domain is modified to exhibit lesser or no affinity for FcRs [26,28,67,68]. Conversely, anti-CTLA-4 constructs with enhanced FcR binding affinity exhibit enhanced anti-tumor activity and Treg depletion [26,28,63]. 

Although anti-CTLA-4 alpaca heavy chain-only Fab could sufficiently block CD80/86-CTLA-4 interactions and localize in B16-F10 melanoma tumors, it did not reduce tumor burden or deplete intra-tumoral Tregs. On the contrary, IgG2a Fc domain conjugation to the Fab fragment reduced tumor burden, depleted intra-tumoral Tregs, and expanded CD4^+^ effector T-cells. However, the anti-tumor response was significantly reduced when a mutation was induced in this full-sized antibody to generate a low-affinity construct for CTLA-4. This observation suggests that Fc-dependent effector functions play a central role in the therapeutic efficacy of anti-CTLA-4 antibodies, while the blocking function of anti-CTLA-4 contributes minimally to its therapeutic efficacy [54,68]. 

While Treg depletion and the consequent increase of the T_effector_: Treg ratio are associated with the clinical response to anti-CTLA-4, some studies have observed an expansion of intra-tumoral and peripheral Tregs following treatment. Analysis of tumor biopsies from stage-matched untreated and ipilimumab or tremelimumab treated patients with bladder cancer, prostate cancer, and metastatic melanoma revealed that patients treated with anti-CTLA-4 agents did not show a decrease of intra-tumoral FOXP3^+^ T-cells [69,70]. While ipilimumab and tremelimumab increased intra-tumoral CD4^+^ and CD8^+^ effector T-cells in these solid malignancies, intra-tumoral FOXP3^+^ T-cells were also increased in melanoma samples. This increase of FOXP3^+^ T-cells was not associated with a decrease of CD68^+^ macrophages responsible for ADCC/P; however, expression of FcRs was not measured [70]. Both ipilimumab and tremelimumab increased intra-tumoral Tregs despite having differing capacities in inducing ADCC/P due to isotype differences. These findings suggest that Treg depletion is not involved in anti-tumor responses induced by anti-CTLA-4 but rather through increasing effector T-cell activation and infiltration.

In addition, an increase of Tregs in PBMCs was observed in pancreatic cancer, metastatic renal cell carcinoma, and metastatic melanoma patients following anti-CTLA-4 treatment. These PBMC-derived Tregs maintained their suppressive function in vitro while activation markers CD25, CD49, HLA-DR, and CD45RO were increased on effector T-cells [71,72,73]. In colorectal cancer mouse models, anti-CTLA-4 depletes intra-tumoral Tregs but enhances the proliferation and number of Tregs in the spleen, non-draining lymph nodes, and tumor-draining lymph nodes [53,71]. The conflicting evidence of Treg depletion versus expansion in response to anti-CTLA-4 may be due to the disruption of the CTLA-4/CD28-dependent feedback loop used by Tregs to regulate its expansion and the heterogenous expression of CTLA-4 on Tregs.

Tregs regulate their local proliferation through stabilized interactions with cognate antigen-presenting dendritic cells (DCs) using the CTLA-4/CD28 axis. Tregs can expand through TCR engagement with CD28 costimulation like conventional T-cells. Tregs use CTLA-4 to downregulate CD80/86 expression on DCs to destabilize cell-cell interactions. CTLA-4 deficiency and anti-CTLA-4 disrupt this rheostat and induce CD28-mediated localized Treg expansion [74,75]. However, the lack of Treg depletion seen in this study was due to the use of anti-CTLA-4 clone 4F10, which has lower Treg depleting capabilities than 9D9 and 9H10 clones [63]. In addition, higher CTLA-4 expression on intra-tumoral Tregs than peripheral Tregs localizes the effects of anti-CTLA-4 in tumors [25,54,76]. Evidence of anti-CTLA-4-mediated Treg expansion has primarily been observed in extra-tumoral tissues. Preferential expansion of Tregs may result from limited anti-CTLA-4 localization and the expression of FcRs in the periphery to facilitate ADCC/P [28]. Nonetheless, ADCC/P-mediated intra-tumoral Treg depletion by anti-CTLA-4 has been strongly supported in vivo and in vitro. The paradoxical nature of anti-CTLA-4 on Tregs highlights the importance of localizing the effects of anti-CTLA-4 in tumors to minimize peripheral effects that contribute to the development of irAEs.

## 5. Alteration of Treg Metabolism and Plasticity

The field of immunometabolism is a new and growing area in immuno-oncology research as ICIs can alter the metabolic profile of T-cells, thereby affecting their function. Effector T-cells primarily use glycolysis to produce energy for T-cell activation and cytokine production [77,78]. In contrast, Tregs use oxidative phosphorylation (OXPHOS) and fatty acid oxidation [78,79]. Tregs are highly plastic cells that can adopt Th1, Th2, and Th17 cell characteristics by expressing the corresponding transcriptional regulators. Metabolism can shape the plasticity of Tregs by affecting the expression of these regulators [79]. Tumors can alter the metabolic environment to exhaust tumor-reactive effector T-cells while enhancing the suppressive capacity of Tregs [80]. Cancer cells deprive the TME of glucose and increase lactic acid production through aerobic glycolysis to support high rates of cell division, known as the Warburg effect. At the same time, Tregs use the lactic acid in the TME to produce pyruvate for OXPHOS and gluconeogenesis such that glucose import is not needed to support cell division [81]. FOXP3 enables Tregs to thrive and maintain their suppressive functions in low-glucose and high-lactic acid environments by increasing lactic acid transport, OXPHOS, and suppressing glycolysis [82,83,84,85]. Impairment of OXPHOS or increased glycolysis can decrease the expression of FOXP3 and alter the suppressive capacity and phenotype of Tregs [81,83,86,87].

Recently, Zappasodi et al. demonstrated that anti-CTLA-4 functionally and phenotypically destabilized Tregs in glycolysis-defective tumors and enhanced anti-tumor immune responses. This tumor phenotype increases the availability of glucose in the TME that can support immune cell infiltration. Anti-CTLA-4 increased effector T-cell infiltration into glycolysis-defective 4T1 murine triple-negative breast tumors and induced a robust immunological memory response. In contrast, glycolysis-proficient 4T1 tumors had less effector T-cell infiltration, and no memory response was established with anti-CTLA-4 therapy. CTLA-4 blockade in glycolysis-defective tumors decreased the expression of CD25 and/or CTLA-4 on intra-tumoral Tregs without affecting FOXP3 expression. In addition, CTLA-4 blockade increased glucose uptake in Tregs through CD28 costimulation, resulting in an increase of IFN-γ and/or TNF-producing Th1-like Tregs. CTLA-4 expression allows Tregs to maintain phenotypic and functional stability in high-glucose environments by inhibiting CD28 costimulation. Given the appropriate conditions in the TME, it suggests that anti-CTLA-4 can promote anti-tumor immune responses by increasing Th1-like Tregs that positively correlate with granzyme B and/or IFN-γ producing CD8^+^ TILs [83]. Employing glycolysis inhibitors may be a strategy to improve the efficacy of anti-CTLA-4 by increasing the plasticity of Tregs in tumors (Figure 3).

Metabolic alteration of Tregs may also induce Th17-like Tregs where IL17^+^FOXP3^+^CD4^+^ T-cells have been identified in the TME [88,89,90,91,92]. Th17 cells are the primary producers of IL-17 and are involved in autoimmune conditions. Blocking CD80/86-CTLA-4 interactions promote Th17 differentiation and autoimmune disease development [93]. Th17 cells are similar to Tregs in terms of plasticity as they can acquire Treg-like characteristics such as the expression of CTLA-4, IL-10, TGF-β, CD25, and FOXP3 [88,90,94]. FOXP3 expression will promote Treg differentiation by suppressing RORγt expression unless IL-6 is present to overcome this suppression to promote Th17 differentiation [95]. In addition to T-cell differentiating cytokines, metabolic fate is critical for development and differentiation of Tregs and Th17 cells. Th17 differentiation requires increasing glycolysis through mTORC signaling and HIF1α expression. In comparison, FOXP3 expression in Tregs suppresses mTORC activation and HIF1α expression in preference of OXPHOS (Figure 3) [89]. 

Th17 cells have been identified in the periphery and TME of ovarian, breast, colorectal, lung, gastric, and pancreatic cancer patients; however, their role in cancer progression is contradictory [96,97,98,99,100,101,102]. The presence of Th17 cells in the TME of breast, colorectal, gastric and pancreatic cancers is negatively correlated with disease progression and patient survival. In contrast, a positive correlation is observed in ovarian and lung cancers [96,97,98,99,100,101]. Th17 cells can promote tumorigenesis by enhancing the effects of vascular endothelial-growth factors on neoangiogenesis and inhibiting CD8^+^ T-cell tumor infiltration [103]. On the other hand, Th17 cells in ovarian and melanoma tumors can enhance anti-tumor immune responses by producing CCL20, CXCL9, and CXCL10 chemokines that promote NK cell, DC, and effector T-cell infiltration [96,104]. It is not known whether anti-CTLA-4 can metabolically interconvert the phenotype of Tregs and Th17 cells and how this plays a role in anti-tumor immunity. Anti-CTLA-4 can potentially act on indoleamine 2,3-dioxygenase (IDO) that is involved in regulating Treg and Th17 cell differentiation. Tregs can increase IDO expression in DCs through the CD80/86-CTLA-4 axis. IDO produces kynurenine and tryptophan metabolites that promote the induction of tolerogenic DCs and Treg differentiation [105,106]. Inhibition of IDO activity could reprogram Tregs into IL-17-producing Th17-like Tregs in the presence of IL-6 [107].

Th17 cells can be expanded ex vivo with anti-CTLA-4 from CD4^+^ T-cells and whole PBMCs isolated from anti-CTLA-4-treated metastatic melanoma patients [93,108]. However, high levels of IL-17 or Th17 cells in the peripheral blood were not correlated with response to immunotherapy but were correlated with irAEs [108,109]. Nevertheless, the adoptive transfer of Th17 cells in pulmonary/subcutaneous melanoma mouse models suppresses tumor growth and enhances the recruitment and activation of effector T-cells [104,110]. Th17 cells play a paradoxical role in cancer development where its effects may be tumor and model-specific. Understanding the Treg/Th17 axis in tumor development may provide an opportunity to either enhance or abrogate Th17 functions in Tregs to improve the efficacy and side effects of anti-CTLA-4.

## 6. Clinical Applications

There are currently 821 clinical trials involving ipilimumab and 210 trials involving tremelimumab in various cancers (https://www.clinicaltrials.gov, accessed on 21 February 2022). To date, ipilimumab monotherapy is only approved for unresectable/metastatic melanoma patients and as adjuvant therapy for patients with resectable melanoma [111]. Ipilimumab is frequently combined with anti-PD-1 antibodies as several clinical trials in various cancers have shown superior efficacy for dual checkpoint inhibition than single-agent treatment. Ipilimumab in combination with nivolumab has been approved for metastatic melanoma, renal cell carcinoma, microsatellite instability-high/mismatch repair-deficient (MSI-H/dMMR) metastatic colorectal cancer, hepatocellular carcinoma, non-small cell lung cancer (NSCLC), and pleural mesothelioma (Table 2) [112,113].

PD-1 is an immune checkpoint molecule expressed on activated T-cells. Upon binding to PD-L1 expressed on hematopoietic and non-hematopoietic cells, PD-1 inhibits effector T-cell functions by attenuating TCR signal transduction. In addition, cancer cells can express PD-L1 to evade host anti-tumor immune responses by inducing exhaustion in tumor-specific effector T-cells [121]. Anti-CTLA-4 enhances T-cell priming, whereas anti-PD-1 reinvigorates exhausted effector T-cells. Combination therapy can increase the activation and infiltration of tumor-specific effector T-cells to maximize anti-tumor immune responses. Notably, the combination of anti-CTLA-4 and anti-PD-1 induces a unique cellular response distinct from the immune response induced by either therapeutic alone [122,123]. In murine MC38 tumors, activated terminally differentiated CD8^+^ T-cells were increased with anti-CTLA-4 and anti-PD-1 combination therapy, but not with anti-PD-1 monotherapy. In addition, combination therapy further increased the frequency of Th1-like CD4^+^ effector T-cells compared to anti-CTLA-4 monotherapy, while single-agent anti-PD-1 had no effect. In melanoma patients, both ipilimumab and nivolumab monotherapy readily expand activated PD-1^+^ CD8^+^ cells in the peripheral blood; however, combination therapy expanded both PD-1^+^CD8^+^ cells and resting PD-1^-^CD8^+^ cells [122]. These findings highlight the dynamic interactions of anti-CTLA-4 and anti-PD-1 in combination therapy that contribute to enhanced therapeutic efficacy.

Adjuvant immunotherapy can activate and expand tumor-specific effector T-cells. After primary cancer treatment(s), these circulating T-cells can eliminate remaining cancer cells and prevent cancer recurrence. Similarly, neoadjuvant immunotherapy can promote anti-tumor immune responses to improve the efficacy of primary interventions and prevent cancer recurrence. Currently, ipilimumab adjuvant monotherapy is approved for resectable stage III melanoma patients. At the same time, anti-CTLA-4 plus anti-PD-1 combination therapy is currently in clinical trials in adjuvant and neoadjuvant settings for several cancers. In a presurgical phase Ib clinical trial, melanoma patients with macroscopic stage III disease were treated with two courses of neoadjuvant ipilimumab (3 mg/kg) plus nivolumab (1 mg/kg) followed by two courses of this combination post-surgery. Those patients had a significant expansion of tumor-resident T-cell clones than patients who received a four-course adjuvant regimen. Based on immunophenotyping, neoadjuvant therapy may be superior in generating a more robust and long-lasting anti-tumor immune response. However, neoadjuvant therapy also induced greater toxicity in patients than adjuvant therapy [124].

Currently, 50% of operable NSCLC patients will have cancer recurrence after surgical resection. In addition, neoadjuvant and adjuvant chemotherapy provide minimal improvement in overall survival, which calls for alternative treatment strategies. The historical major pathological response (MPR) rate is ~15% in neoadjuvant chemotherapy treated NSCLC patients. The neoadjuvant ipilimumab (1 mg/kg) plus nivolumab (3 mg/kg) regimen in the phase II randomized NEOSTAR clinical trial resulted in an MPR rate of 50% compared to 24% in the single-agent nivolumab neoadjuvant arm. Ipilimumab plus nivolumab increased the frequency of CD3^+^, CD3^+^CD8^+^, and CD3^+^CD8^+^CD45RO^+^ memory TILs in post-operative tumor samples compared to preoperative samples, whereas nivolumab monotherapy had no effect [125].

Similarly, patients with operable high-risk urothelial carcinoma ineligible for cisplatin neoadjuvant therapy do not have an alternative treatment option. Recently, a pilot study treating patients with two courses of tremelimumab (75 mg/kg) and durvalumab (1500 mg/kg) every four weeks prior to cystectomy resulted in a complete pathological response in 37.5% of patients. Among those that received surgery, 58% had downstaging of pT1 or less [126]. Current trials using anti-CTLA-4 and anti-PD-1 as neoadjuvant therapy have shown promising results; however, further studies are required to refine dosing regimens and drug combinations to minimize toxicity.

Despite the low response rates as a single-agent and relatively high occurrences of irAEs associated with anti-CTLA-4, it has a high potential in treating immunologically-cold tumors. These tumors lack immune cell infiltration and often do not respond well to ICIs compared to immunologically-hot tumors [31]. Anti-PD-1 therapy cannot stimulate a robust anti-tumor immune response unless pre-existing tumor-specific effector T-cells are present [127]. Anti-CTLA-4, on the other hand, can increase T-cell activation while inhibiting/depleting intra-tumoral Tregs to promote immune cell infiltration. The efficacy of anti-CTLA-4 is therefore dependent on the accessibility and array of neoantigens [128,129]. Clinical response to anti-CTLA-4 in advanced melanoma is associated with tumor mutational load, and an increase of TILs can be observed as early as three weeks post-treatment [128,130]. A recent study in anti-PD-1/L1-refractory patients has shown promising results using a low dose of ipilimumab (1 mg/kg) once every three weeks for a total of four doses in combination with pembrolizumab [111]. These patients had an objective response rate of 29%, a median progression-free survival of 5.0 months, and a median overall survival of 24.7 months. Notably, many patients with PD-L1-negative and immune-cold melanoma tumors were among the responders in this study. These findings further highlight the role of the anti-CTLA-4 blockade in activating immune cells in peripheral immune organs followed by the influx of activated T-cells into immune-cold tumors that are commonly refractory to anti-PD-1 [73,131].

Radiotherapy has been demonstrated to improve response to ICIs in preclinical and clinical studies. Radiotherapy can induce DNA damage in tumor cells and increase the production of neoantigens and damage-associated molecular patterns (DAMPs) that promote immune cell activation [132,133]. Increased neoantigen production from radiotherapy can broaden the repertoire of anti-tumor T-cells to which anti-CTLA-4 can act in tandem to promote T-cell activation and expansion [134,135]. Radiotherapy dose-dependently sensitized low mutational burden Kras^G12D^ x p53^−/−^ sarcoma lines to anti-CTLA-4 plus anti-PD-1 by increasing its immunogenicity through radiation-induced neoantigen production [132]. Similarly, anti-CTLA-4 in combination with radiotherapy in an immunologically-cold 4T1 breast cancer mouse model increased overall survival and decreased lung metastasis, whereas either treatment alone was not effective [136,137]. In a phase II clinical trial, 18% of chemo-refractory metastatic NSCLC patients treated with focal radiation and anti-CTLA-4 exhibited an abscopal response. Responders had an increase of systemic IFN-β and diversification and expansion of tumor-specific TCR clones in the peripheral blood compared to baseline levels [138]. These results further support the importance of neoantigens in anti-CTLA-4 response and the potential use of radiotherapy to enhance anti-CTLA-4 efficacy in patients with immunologically-cold and/or low mutational burden tumors.

## 7. Next-Generation CTLA-4 Targeting and Strategies to Improve Therapeutic Efficacy

Tremelimumab is another anti-CTLA-4 antibody that has been in several clinical trials for the treatment of solid cancers. Tremelimumab and ipilimumab share the same binding site on the CTLA-4 molecule with similar binding affinity [58]. However, tremelimumab has not received FDA approval as it has not met the primary endpoints in clinical trials. In a phase III randomized clinical trial, tremelimumab failed to improve overall survival in patients with metastatic melanoma compared to patients receiving standard-of-care chemotherapy at the time of the study. The median overall survival was 12.6 months for tremelimumab and 10.7 months for chemotherapy [139]. In comparison, overall survival was significantly improved in patients treated with ipilimumab plus dacarbazine (11.2 months) compared to dacarbazine plus placebo (9.1 months) [140]. However, a retrospective study on 143 patients with metastatic melanoma previously treated with tremelimumab in phase I and II clinical trials had an objective response rate of 15.6% with a median duration of response of 6.5 years and an estimated 12.5-year survival rate of 16% [141]. Based on these findings, despite the lack of efficacy reported in trials, tremelimumab could provide long-term benefits in responders even with a single dose [141,142]. Although the occurrence of grade 3 or 4 adverse events was significantly higher in patients treated with ipilimumab plus dacarbazine, the 3-year survival rate was higher (20.8%) than dacarbazine plus placebo (12.2%) [140]. Compared to ipilimumab, the risk of developing grade 1–5 or grade 3 and 4 adverse events is lower with tremelimumab in melanoma and lung cancer patients [143]. Tremelimumab is an IgG2 antibody with less affinity towards FcγRs than ipilimumab with an IgG1 isotype [144]. It has yet to be determined how the reduced Fc-dependent effector functions of an IgG2 isotype directly contribute to tremelimumab’s toxicity and therapeutic efficacy. A recent report of tremelimumab plus durvalumab treatment in patients with unresectable hepatocellular carcinoma showed that a single dose (300 mg) of tremelimumab with durvalumab (1500 mg) followed by durvalumab once every 4-weeks had superior efficacy compared to either as monotherapy with repeated dosing [145]. Single-dose tremelimumab plus durvalumab yielded the highest overall response rate (24%) and overall survival (18.73 months) with reduced toxicity compared to other treatment regimens and sufficiently expanded CD8^+^ T-cells. The benefit-risk profile of single-dose tremelimumab plus durvalumab suggests that higher or repeated dosing of anti-CTLA-4 that is often associated with toxicity may not be required to achieve a clinical effect [145]. Despite ipilimumab’s potential value in transforming the immune phenotype of low tumor mutation burden and immune-cold tumors, higher rates of irAEs can limit its use. Strategies such as changing the dosing schedule or dosage administration of ipilimumab paired with reverse translational studies to confirm the immune mechanisms of the drug should be considered in designing future clinical trials.

A new anti-CTLA-4 antibody, quavonlimab, is currently in phase I clinical trials with pembrolizumab in patients with advanced solid tumors. First-line treatment in patients with advanced NSCLC with quavonlimab plus pembrolizumab resulted in a median overall survival of 16.5 months and a 1-year overall survival rate of 67%. Patients that received 25 mg quavonlimab every six weeks with 200 mg pembrolizumab every three weeks had an objective response rate of 37% [146]. Similarly, a study with ipilimumab plus nivolumab in patients with NSCLC with ≥1% PD-L1 expression had an objective response rate of 35.9%, a median overall survival of 17.1 months, and a 1-year overall survival rate of 62.6% [118]. In addition, the dosage of quavonlimab was associated with an increase of proliferating CD4^+^ and CD8^+^ T-cells in the peripheral blood, and the total number of circulating CD4^+^ T-cells was correlated with objective response rates [146].

High expression of CTLA-4 on tumor-infiltrating Tregs relative to peripheral Tregs promotes the localization of anti-CTLA-4 in tumors [28]. However, anti-CTLA-4 can elicit autoimmune symptoms by affecting peripheral Tregs. irAEs associated with anti-CTLA-4 are correlated with systemic T-cell activation and expansion of self-reactive T-cells [147]. Strategies that aim to target intra-tumoral Tregs preferentially with anti-CTLA-4 can enhance anti-tumor immune responses while minimizing toxicity. In addition to CTLA-4, the expression of Glucocorticoid-Induced TNFR-Related protein (GITR), OX40, and CD25 are higher on intra-tumoral Tregs than peripheral Tregs [28]. Due to the differential expression of these markers in intra-tumoral and peripheral compartments, dual targeting on Tregs can enhance anti-tumor immune responses and the efficiency of Treg inhibition/depletion. Both GITR and OX40 are expressed on effector T-cells and Tregs. On effector T-cells, engagement of GITR and OX40 to their respective ligands provides costimulatory signaling that promotes proliferation and enhances effector functions. Whereas engagement of GITR and OX40 on Tregs can abrogate their suppressive functions, prevent Treg differentiation, and decrease the expression of FOXP3 [148,149]. Therefore, a robust anti-tumor effect can be induced by acting on both the immunosuppressive and effector compartments of immune responses. In mice models, agnostic anti-GITR and anti-OX40 were shown to reduce tumor burden, deplete intra-tumoral Tregs, and increase the infiltration and function of effector T-cells [150,151,152,153]. Combination therapy of anti-CTLA-4 with anti-GITR or anti-OX40 agonists induces a more robust anti-tumor immune response than any of these as single-agent therapies [154,155,156]. Similarly, CTLA-4 blockade with anti-CD25-mediated Treg depletion results in maximal rejection of B16-F10 melanoma tumors [157].

As discussed, anti-CTLA-4-mediated Treg depletion is limited by the expression of FcRs on innate cells and the efficiency of ADCC/P. Recently, Zhang et al. generated a bispecific anti-CTLA-4xSIRPα antibody with enhanced targeting of intra-tumoral Tregs and ADCP-mediated Treg depletion. CD47 expressed on hematopoietic cells can engage with SIRPα on phagocytes to inhibit ADCP signaling events. In mice and patient samples, CD47 and CTLA-4 co-expression are the highest amongst intra-tumoral Tregs compared to peripheral Tregs and intra-tumoral effector T-cells. Anti-CTLA-4xSIRPα enhances ADCP-mediated Treg depletion by simultaneously blocking the “do not eat me” signaling events from CD47-SIRPα. Although the anti-CTLA-4xSIRPα construct has less binding affinity toward CTLA-4, it was more effective than anti-CTLA-4 plus anti-CD47 combination treatment in depleting highly immunosuppressive ICOS^high^ intra-tumoral Tregs and reducing tumor burden in colorectal cancer mouse models. Furthermore, anti-CTLA-4xSIRPα exhibited less toxicity than anti-CTLA-4 or anti-CD47 in humanized mouse models [67]. This suggests that the safety of anti-CTLA-4 may be improved by reducing its affinity towards CTLA-4 without compromising therapeutic efficacy.

Several studies have also demonstrated that anti-CTLA-4 constructs lacking blocking activity have reduced toxicity in humanized animal models [53,54,147]. Du et al. and Stone et al. synthesized humanized antibodies that lacked CD80/86-CTLA-4 blocking activity but had enhanced FcR activity. The anti-tumor response induced by these antibodies and their ability to deplete intra-tumoral Tregs was superior to ipilimumab in human *CTLA4* knock-in mouse models. Stone et al. described an anti-CTLA-4 (GIGA-564) antibody that efficiently depletes intra-tumoral Tregs in MC38 tumors and exhibited more murine FcγRIV signaling than ipilimumab and in the human FcγRIIIa. Consistent with irAEs reported in humans, ipilimumab plus pembrolizumab combination therapy induced colitis and skin inflammation in human *CTLA4* and *PD1* double knock-in mice. However, this was not observed with GIGA-564 plus pembrolizumab combination therapy [53]. Unlike pH-insensitive GIGA-564, Zhang et al. generated pH-sensitive HL12 and HL32 that also exhibited less toxicity and enhanced ADCC activity than ipilimumab. The low pH found in endosomes promotes the dissociation of HL-12 and HL-13 from CTLA-4, which allows for both CTLA-4 and anti-CTLA-4 to be recycled [158].

In contrast, ipilimumab downregulates cell surface expression of CTLA-4 by inhibiting LRBA-dependent recycling that is associated with the development of irAEs. An increase of CTLA-4 expression on the surface of Tregs and increased bioavailability of anti-CTLA-4 can enhance ADCC/P activity. Impairment of the LRBA-dependent recycling pathway abolished the enhanced ADCC activity exhibited by HL-12 and HL-32 [158]. More importantly, maintaining CTLA-4 expression on peripheral Tregs can preserve immunoregulatory activities necessary to prevent irAEs in the presence of anti-CTLA-4. Although higher dosages of anti-CTLA-4 can confer better clinical outcomes, the development of irAEs is often the limiting factor. These data suggest that blocking CD80/86-CTLA-4 interactions may not be necessary, and next-generation anti-CTLA-4 antibodies that enhance ADCC/P on Tregs may be safer and have better therapeutic efficacy [147].

## 8. Conclusions

Despite its toxicity, anti-CTLA-4 has regained interest in recent years due to its unique immunological activity in hard-to-treat cancers, especially as a combination agent with anti-PD-1. Efforts have been made to decipher fundamental mechanisms of existing anti-CTLA-4 therapeutics to create a new generation of therapeutics with minimal toxicity and improved efficacy. Tregs in the TME remain an obstacle to effector T-cell priming and the exertion of anti-tumor immune mechanisms. Research to date suggests that enhancing anti-CTLA-4-mediated Treg depletion may be the foremost strategy for developing next-generation anti-CTLA-4 antibodies.

## Figures and Tables

**Figure 1 cancers-14-01580-f001:**
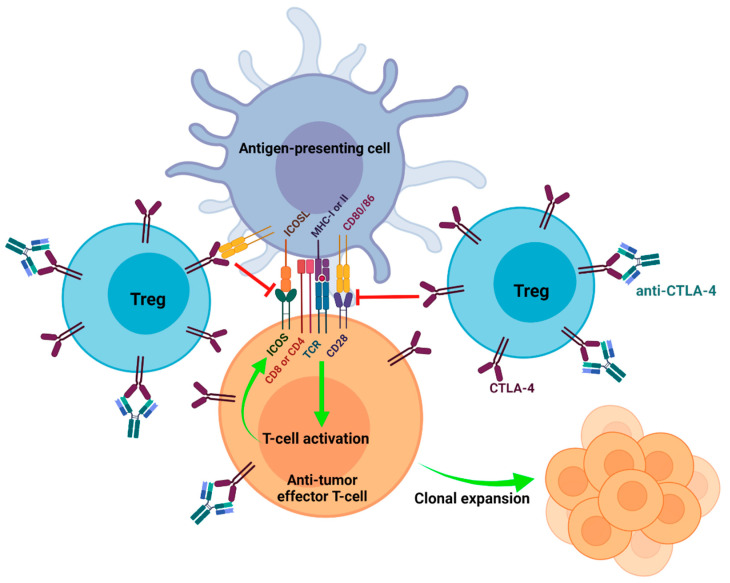
Enhancing the priming of effector T-cells by blockading CD80/86-CTLA-4 interactions. High expression of CTLA-4 on Tregs contributes to their immunosuppressive phenotype. Effector T-cells can express CTLA-4 transiently after T-cell activation. CTLA-4 engagement with CD80/86 on antigen-presenting cells inhibits CD28 costimulation that is required for T-cell activation and the upregulation of ICOS. Anti-CTLA-4 binds to CTLA-4 and inhibits CD80/86-CTLA-4 interactions to increase the activation of anti-tumor effector T-cells. T-cell activation results in clonal expansion and the employment of effector mechanisms that facilitate anti-tumor immune responses.

**Figure 2 cancers-14-01580-f002:**
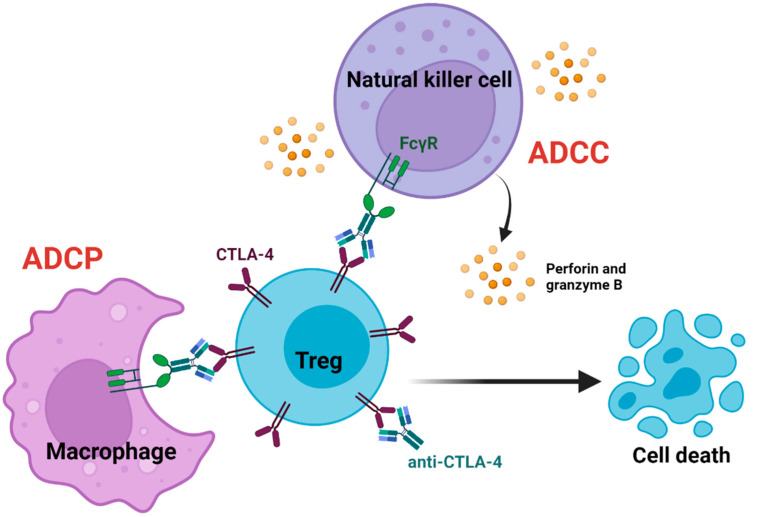
Antibody-mediated Treg depletion. Anti-CTLA-4 bound to Tregs can engage with FcγRs expressed on innate cells to deplete Tregs. Macrophages and natural killer cells can deplete Tregs through antibody-dependent cellular phagocytosis (ADCP) and antibody-dependent cellular cytotoxicity (ADCC). Depleting intra-tumoral Tregs promotes anti-tumor immune responses by transforming the immunosuppressive nature of the tumor microenvironment into a pro-inflammatory microenvironment. This is enabled by indirectly increasing anti-tumor effector T-cells’ activation, infiltration, and effector functions.

**Figure 3 cancers-14-01580-f003:**
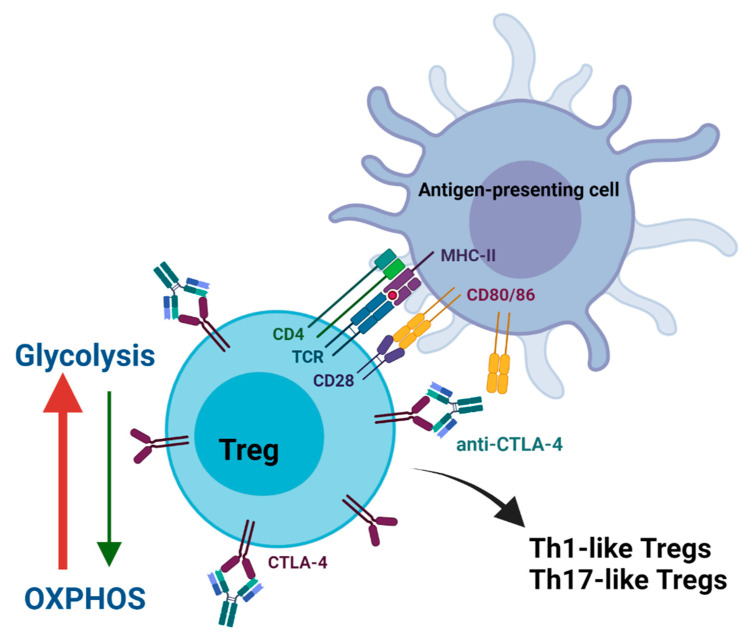
Alteration of Treg metabolism and plasticity. Anti-CTLA-4 allows CD28 on Tregs to engage with CD80/86 on antigen-presenting cells. Costimulatory signaling shifts the metabolism of Tregs from oxidative phosphorylation (OXPHOS) to glycolysis. Increasing glycolysis can functionally and phenotypically destabilize the immunosuppressive nature of Tregs. Tregs can adopt the pro-inflammatory characteristics of Th1 and Th17 cells to contribute to anti-tumor immune responses.

**Table 1 cancers-14-01580-t001:** Summary of common anti-CTLA-4 antibodies.

	Drug	Isotype
Human	Ipilimumab	Human IgG1
	Tremelimumab	Human IgG2
Mouse	9H10	Syrian Hamster IgG
	9D9	Mouse IgG2b
	4F10	Armenian Hamster IgG

**Table 2 cancers-14-01580-t002:** Summary of clinical trials for cancers with FDA-approved use of ipilimumab.

Trial	Cancer	Treatments	Overall Response Rate (%) ^1^	Overall Survival Rate (%) ^2^	
NCT00094653 [2]	Unresectable or metastatic melanoma	gp100 (*n* = 136)	1.5	13.7 (2-yr)	
		Ipilimumab (3 mg/kg) (*n* = 137)	10.9	23.5	
		Ipilimumab (3 mg/kg) + gp100 (*n* = 403)	5.7	21.6	
NCT01844505 [114]	Unresectable or metastatic melanoma (*BRAF* V600—wildtype or mutant)			*BRAF*-wildtype	*BRAF*-mutant
		Nivolumab (3 mg/kg) (*n*-316)	45	22 (6.5-yr)	25 (6.5-yr)
		Ipilimumab (3 mg/kg) (*n* = 315)	19	42	43
		Ipilimumab (3 mg/kg) + nivolumab (3 mg/kg) (*n* = 314)	58	46	57
NCT00636168 [3]	Stage III melanoma—adjuvant therapy	Placebo (*n* = 476)	N/A	54.4 (5-yr)	
		Ipilimumab (10 mg/kg) (*n* = 475)		65.4	
NCT02231749 [115]	Advanced renal cell carcinoma	Sunitinib (*n* = 422)	26.5	60 (1.5-yr)	
		Nivolumab (3 mg/kg) + ipilimumab (1 mg/kg) (*n* = 425)	41.6	75	
NCT02231749 [116]	MSI-H/dMMR metastatic colorectal cancer	Nivolumab (3 mg/kg) + ipilimumab (1 mg/kg) (*n* = 119)	55	85 (1-yr)	
NCT01658878 [117]	Hepatocellular carcinoma	Nivolumab (1 mg/kg) + ipilimumab (3 mg/kg) (*n* = 50)	32	48 (2-yr)	
		Nivolumab (3 mg/kg) + ipilimumab (1 mg/kg) (*n* = 49)	31	30	
		Nivolumab (3 mg/kg) + ipilimumab (1 mg/kg) (*n* = 49)	31	42	
NCT02477826 [118]	Metastatic NSCLC (≥1% PD-L1)	Nivolumab (3 mg/kg) + ipilimumab (1 mg/kg) (*n* = 396)	35.9	40 (2-yr)	
		Platinum-doublet chemotherapy (*n* = 397)	30	32.8	
NCT03215706 [119]	Metastatic or recurrent NSCLC	Nivolumab (360 mg) + ipilimumab (1 mg/kg) + platinum-doublet chemotherapy (*n* = 361)	38	63 (1-yr)	
		Platinum-doublet chemotherapy (*n* = 358)	25	47	
NCT02899299 [120]	Unresectable malignant pleural mesothelioma	Nivolumab (3 mg/kg) + ipilimumab (1 mg/kg) (*n* = 303)	40	41 (2-yr)	
		Platinum + pemetrexed chemotherapy (*n* = 302)	43	27	

^1^ Values based on blinded independent central review (BICR) assessment if available. ^2^ Overall survival rate timepoint indicated in brackets.

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
