# Peer review of "Addressing the Elephant in the Immunotherapy Room: Effector T-Cell Priming versus Depletion of Regulatory T-Cells by Anti-CTLA-4 Therapy"

_cancers, 2022, doi:10.3390/cancers14061580_

Round 1
Reviewer 1 Report
Reviewer comments:
Comments to the Author
The present review discusses proposed fundamental mechanisms of anti-CTLA-4 therapy, novel uses of anti-CTLA-4 in cancer treatment and approaches to improve the therapeutic efficacy of anti-CTLA-4. However, the potency of using anti-CTLA-4 in cancer patients are relatively low, and a high percentage of patients experience severe immune-related adverse events. Authors discussed about the Treg metabolism and plasticity and how the understanding of Treg/Th17 axis in tumor development may provide an opportunity to either enhance or abrogate Th17 functions in Tregs to improve the efficacy and side effects of anti-CTLA-4.
This review is for the most part well written and discussed the recent studies. However to increase the readability, authors are advised to incorporate some suggestions.
Minor criticisms
- Authors should provide a comprehensive table to show the role or effect of anti-CTLA-4 response on different types of cancer prognosis including references.
- Please provide a table or a figure including all the drugs which are discussed in the “Blockading CD80/86-CTLA-4 interactions” section.
- Authors are advised to include a section regarding radiotherapy, which are applied to induce anti-tumor T cells in lymphocyte-poor tumors, and possibly benefit patients who would otherwise fail to respond to immune checkpoint inhibitors.
Author Response
This review is for the most part well written and discussed the recent studies. However to increase the readability, authors are advised to incorporate some suggestions.
Minor criticisms
- Authors should provide a comprehensive table to show the role or effect of anti-CTLA-4 response on different types of cancer prognosis including references.
We thank the reviewer for their insight and suggestions. We have provided a table summarizing results from clinical trials using anti-CTLA-4 for cancers currently approved for single-agent ipilimumab or in combination with anti-PD1 (line 400-402).
- Please provide a table or a figure including all the drugs which are discussed in the “Blockading CD80/86-CTLA-4 interactions” section.
A table summarising current anti-CTLA-4 therapeutics used in humans and mouse anti-CTLA-4 clones have been included (line 124-132).
- Authors are advised to include a section regarding radiotherapy, which are applied to induce anti-tumor T cells in lymphocyte-poor tumors, and possibly benefit patients who would otherwise fail to respond to immune checkpoint inhibitors.
A paragraph describing the use of radiotherapy and anti-CTLA-4 to treat immunologically-cold tumors was included in the “Clinical applications” section (line 474-491).
Reviewer 2 Report
T-cell activation is currently considered the primary mechanism of action of anti-tumour therapies, targeting cytotoxic T-lymphocyte-associated protein 4 (CTLA-4), an immune checkpoint molecule. However, growing evidence in the literature suggests that targeting intra-tumour regulatory T cells (Tregs) is the primary mechanism of action of these agents. Indeed, CTLA-4 is overexpressed on Tregs capable of inhibiting the activation of effector T cells (Teff). Anti-CTLA-4 therapies would therefore prevent this inhibition.
The review summarises current knowledge of the function of CTLA-4 as an immune checkpoint. It details the mechanisms by which this protein inhibits Teff activation and explains the basic mechanisms proposed for anti-CTLA-4 therapies.
Given the importance of CTLA-4 in current and developing targeted therapies, there are obviously several reviews on this topic, but this review is complementary because it does not treat the subject in exactly the same way.
The review is well detailed, the rationale is easy to follow and the main published works in the literature are cited.
The figures are well annotated and commented, easy to understand.
The authors do not use self-citation.
The review cites 143 references but more than half of the references were published more than 5 years ago.
Line 702: in the references n° 43 and n°50 the dates are missed
Author Response
The figures are well annotated and commented, easy to understand.
The authors do not use self-citation.
We thank the reviewer for their comments.
The review cites 143 references but more than half of the references were published more than 5 years ago.
We thank the reviewer for their comments. Much of the research on anti-CTLA-4 was conducted in the past decade; hence some of the citations were >5 years ago. However, all recent studies on anti-CTLA-4 relevant to the review were included.
Line 702: in the references n° 43 and n°50 the dates are missed
We thank the reviewer for their careful review of our manuscript. References missing dates have been corrected (line 734, 750).
Reviewer 3 Report
This review accurately addresses the role of the CTLA-4 molecule in cancer, describing the mechanism underlying anti-CTLA4 therapy and novel approaches to improve its therapeutic efficacy. The review is detailed and accurate and may be considered for publication in the journal. To make it even clearer and updated, the authors could insert in the paragraph "clinical application" a table showing the various clinical trials in use today, highlighting the number of trials and the characteristics of each.
Author Response
This review accurately addresses the role of the CTLA-4 molecule in cancer, describing the mechanism underlying anti-CTLA4 therapy and novel approaches to improve its therapeutic efficacy. The review is detailed and accurate and may be considered for publication in the journal. To make it even clearer and updated, the authors could insert in the paragraph "clinical application" a table showing the various clinical trials in use today, highlighting the number of trials and the characteristics of each.
We thank the reviewer for their time and suggestions. There are currently >1000 clinical trials involving ipilimumab or tremelimumab, we have provided a reference for information on clinical trials on line 390. A table summarizing results from clinical trials using ipilimumab for cancers currently approved for single-agent ipilimumab or in combination with anti-PD-1 was also provided (line 400-402).